# The Effects of Storage Temperature, Light Illumination, and Low-Temperature Plasma on Fruit Rot and Change in Quality of Postharvest Gannan Navel Oranges

**DOI:** 10.3390/foods11223707

**Published:** 2022-11-18

**Authors:** Ying Sun, Yuanyuan Li, Yu Xu, Yali Sang, Siyi Mei, Chaobin Xu, Xingguo Yu, Taoyu Pan, Chen Cheng, Jun Zhang, Yueming Jiang, Zhiqiang Gao

**Affiliations:** 1College of Life Sciences, Gannan Normal University, Ganzhou 341000, China; 2Guangdong Provincial Key Laboratory of Applied Botany, South China Botanical Garden, Chinese Academy of Sciences, Guangzhou 510650, China

**Keywords:** *Citrus sinensis*, water loss, quality, light, plasma, temperature, storage

## Abstract

Gannan navel orange (*Citrus sinensis* Osbeck cv. Newhall) is an economically important fruit, but postharvest loss occurs easily during storage. In this study, the effects of different temperatures, light illuminations, and low-temperature plasma treatments on the water loss and quality of the Gannan navel orange were investigated. The fruit began to rot after 90 d of storage at 5 °C and 20–45 d at 26 °C. Navel oranges stored at 26 °C had 7.2-fold and 3.1-fold higher rates of water loss at the early and late storage stages, respectively, as compared with those stored at 5 °C. Storage at 5 °C decreased the contents of total soluble solids at the early storage stage and the contents of titratable acids at the late storage stage, whereas storage at 26 °C decreased the contents of total soluble solids at the late storage stage and the contents of titratable acids at the early storage stage, respectively. Application of low-temperature plasma produced by air ionization for 6 min, or continuous blue or red light illumination significantly inhibited water loss within 7 and 21 d of storage at 22 °C, respectively, but exhibited no significant effect on fruit quality. Furthermore, the low-temperature plasma treatment protected against fruit rot. Thus, treatment with low-temperature plasma followed by storage at a low temperature under continuous red or blue light illumination was of potential value as a green technology for preserving Gannan navel orange during storage.

## 1. Introduction

The Gannan navel orange (*Citrus sinensis* Osbeck cv. Newhall), planted in Jiangxi Province in southern China, show high-quality characteristics, including moderate contents of soluble solids and titratable acids with a sweet flavor and rich health-promoting compounds. The fruit is highly favored by consumers, but postharvest rot occurs easily. Furthermore, with increasing planting area and production output, postharvest loss from fruit rot becomes an outstanding problem during storage. About 90% of citrus fruit rot during storage and transport is caused by *Penicillium digitatum* and *Penicillium italicum* [1]. Currently, chemical fungicides, such as prochloraz, imazalil, iminoctadine trialbesilate, and auxin analogue 2, 4-dichlorophenoxyacetic acid (2, 4-D), are usually used to control postharvest rot of Gannan navel oranges during storage [2]. These chemical fungicides mainly inhibit the germination of fungal spores, destroy the integrity of spore cell walls, increase the permeability of cell membranes, and interfere with the metabolism and enzymatic activity of pathogenic microbes [1,3,4,5,6]. Peel is the initial symptom of mold causing postharvest fruit rot. At the early stage of fungal infection, pathogenic factors, such as cell-wall-degrading enzymes, ethylene, organic acids, and effectors, are up-regulated [7,8]. For example, in *Penicillium digitatum*, the pH-signaling transcription factor PdPacC regulates the expression of cell-wall-degrading enzymes [9], which affects the ability of the mold to infect fruit under storage conditions.

Several studies have investigated different physical and chemical treatments to prevent fruit rot. For example, chlorine dioxide, natamycin, and polypeptide sanxiaeptin isolated from *Penicillium oxalate* have been shown to inhibit the growth of green mold and control postharvest decay of navel orange fruit [6,10,11]. It was also reported that a high concentration of CO_2_ or O_2_ [12], heat shock treatment [13,14,15,16,17], and blue light application [18,19] are effective in inhibiting pathogen infection and maintaining the quality of postharvest citrus fruit. Considering the use of chemical fungicides, the development of resistance, the risk of environmental contamination, as well as the possible health hazards, extracts from edible plants, traditional Chinese medicine, and antagonistic yeasts have been investigated as “green” preservatives for citrus fruit [20,21,22]. In recent years, nonchemical postharvest treatments, such as low-temperature plasma and light illumination, have received more attention. The use of these treatments for control of fruit rot and maintenance of the quality of Gannan navel oranges during storage needs to be further investigated.

The objective of this study was to determine the biological effects of temperature, light, and low-temperature plasma on water loss and fruit quality, which are related to postharvest decay of Gannan navel oranges during storage. The results of this study can aid in elucidating the biological basis of physical preservation and to develop a green preservation technology for Gannan navel oranges during transport and storage.

## 2. Materials and Methods

### 2.1. Plant Materials

Gannan navel orange fruits (*Citrus sinensis* Osbeck cv. Newhall) were obtained from an orchard in Xinfeng County, Ganzhou City in November of 2020 and in November 2021, respectively, and used for validation experiments. Fruits of similar color and size (6.5–7.0 cm in diameter and approximately 220 g in weight) without any surface damage were selected (Appendix A) then washed with tap water and finally dried at ambient temperature (about 15 °C) before use in subsequent storage experiments.

### 2.2. Storage Temperature

Experiments were conducted at two temperatures (5 and 26 °C) from November 2020 to June 2021 and at three temperatures (5, 15, and 22 °C) from November 2021 to March 2022 in air-conditioned rooms. Twenty fruits were labeled and weighed per week during storage. These weights were used for calculating the water loss rate for each treatment. Once every 2 weeks during storage, six fruit were sampled from the boxes and used for measurements of fruit firmness and concentrations of total soluble solids (TSS), and titratable acids (TA).

### 2.3. Light Treatment

The orange fruits stored in the air-conditioned room at 22 °C was illuminated with white, red, yellow, green, or blue LED light (12 Watts, 11.0 cm in diameter and irradiation area of approximately 8–15 m^2^). The vertical distance between the fruits and LED light was approximately 30 cm. Twenty fruits were labeled and continuously irradiated for 21 d. During this period, the water loss was determined once every 3 d, and TSS and TA contents were determined after 21 d of treatment.

### 2.4. Low-Temperature Plasma

Low-temperature plasma treatment was conducted by our previously described method [23] using a high-frequency power supply (CTP-2000K) to generate plasma. The low-temperature plasma generator was equipped with an axial flow discharge component with a total length of 0.20 m, a sealed upper end, and a side end with an air inlet. Plasma was collected at the tail end. The dielectric barrier discharge mode was set, while air and oxygen were ionized at an inlet flow rate of 0.02 L s^−1^ (Figure 1).

The low-temperature plasma produced by air ionization was used to treat suspensions of *P. digitatum* and *P. italicum* spores obtained from decayed fruit for 4, 10, and 20 min, respectively. A 1 mL sample of the treated spore suspension was transferred to a 1.5 mL microcentrifuge tube, and then two drops of 0.4% trypan blue staining solution were added. After staining for 2 min, a 10 µL aliquot of sample was placed onto a blood cell counting plate, and the morphology and staining color of these spores were observed using a light microscope (BX41TF, Olympus corporation, Tokyo Japan). The dead cells stained with trypan blue appeared blue in color, while the living cells were not stained.

In addition, low-temperature plasma was used to treat 16 navel orange fruits selected from an air-conditioned room at 5 °C and placed into an acrylic tube (0.20 m in diameter and 0.30 m in height with a 0.04 m circular hole at the upper and lower ends for gas inlet and outlet). Low-temperature plasma produced by ionization of air, and oxygen was fed into the upper gas hole for 3 and 6 min, while the excess gas was vented from the lower gas hole (Figure 1). After low-temperature plasma treatment, water loss rate was calculated by measuring fresh weight once a day in an air-conditioned room at 22 °C.

### 2.5. Fungal Infection

*P. digitatum* and *P. italicum* spores were collected from the surfaces of decayed Newhall navel orange fruit in storage room, mixed with sterile water, filtered through Miracloth (475855-1R, EMD Millipore Corporation, Billerica, MA, USA) into a 50 mL centrifuge tube, and diluted to 2 × 10^5^ spores/mL using sterile water. Spores were counted under a light microscope (BX41TF, Olympus corporation, Tokyo, Japan) using a blood cell counting plate. Two methods were used to test the fungal infection of orange fruits obtained from a Xinfeng orchard and stored in an air-conditioned room at 5 °C. One method was in vivo drilling inoculation of a 10 μL spore suspension on the fruit surface, while the other was noninvasive inoculation in which a 25 μL aliquot of the spore solution was dropped onto a piece of filter paper (0.005 m in diameter after cutting) and then fixed onto the fruit surface using gauze (1530-C, Micropore, Minnesota Mining Manufacturing Medical Equipment Co., Ltd., Shanghai, China). These treated fruits were stored in an air-conditioned room at 26 °C, and the mold growth at the inoculation sites was observed every 3 d after in vivo drilling inoculation and 1 month after noninvasive inoculation.

### 2.6. The Rupture of Pericarp Oil Cells

Peels of 12 navel orange fruit were treated with oils extracted from navel orange exocarp, or pericarp oil cells were artificially ruptured using a syringe needle. Fruits without any treatment were used as a control. The average and accumulated water loss were measured every 3 d in an air-conditioned room at 22 °C.

### 2.7. Firmness

The firmness of navel oranges after temperature and light treatments was measured using a research texture analyzer (TA.XTC-18, Shanghai Baosheng Industrial Development Co., Ltd., Shanghai, China). The deformation force (N) was used as a measure of fruit firmness.

### 2.8. Fruit Quality

After temperature, light, and low-temperature plasma treatments, navel orange fruit were juiced. The TSS content was determined using an ATAGO PAL-1 digital display refractometer, while the TA content was measured using an ATAGO PAL-Easy ACID1 citrus acidity tester.

### 2.9. Statistical Analysis

Experiments were carried out with a random design. All assays in these experiments were conducted with three replications. Data are presented as the means ± standard deviation (SD), and the significant differences were analyzed by the Student’s *t*-test (*p* < 0.05 or *p* < 0.01).

## 3. Results

### 3.1. Fruit Rot

To test the effect of storage temperature on fruit rot, oranges were stored at 5 °C (low temperature) and 26 °C (high temperature). Newhall navel oranges began to rot after 20 d of storage at 26 °C but after 90 d at 5 °C, which showed that low-temperature storage could inhibit pathogen infection and delayed the initiation time of fruit rot (Figure 2). As shown in Figure 2, the resistance of Gannan navel oranges to pathogen infection depended on the storage temperature.

Mechanical damage to the orange peel caused by harvesting and storage is unavoidable and leads to fruit rot. It was observed that decay of fruit in storage at 5 °C was generally initiated by a cutting injury of fruit pedicle during harvest (Figure 3A) or physical damage caused by transport and gravity extrusion after harvest (Figure 3B). To test the role of mechanical damage in the initial stage of infection, we performed invasive and noninvasive inoculation experiments. After one month of storage at 26 °C following noninvasive inoculation, no marked spore germination or fruit rot was observed on the fruits’ surface, with only a few moldy patches appearing (Figure 3C–E). However, within 3 d of drill damage inoculation at 26 °C, the inoculation sites were covered with fungal mycelium. In contrast, there was no pathogen growth in these sites inoculated with sterile water or pathogen spores inactivated by low-temperature plasma treatment (Figure 3H).

Our previous experiment demonstrated that application of low-temperature plasma produced by air ionization could preserve navel oranges by effectively killing pathogenic fungal spores and reducing fruit rot [23]. In this study, after application of low-temperature plasma (oxygen ionization) for 4, 10, and 20 min, the blue spores were observed, but control spores remained white (Figure 4), which indicated that the treatment killed the pathogenic spores.

There is a layer of oil cells on fruit epidermis, and these cells play an important role in fruit rot and water loss. To determine whether the oil from these cells or the rupture of pericarp oil affected fruit rot, we treated the surfaces of fruit using the oil extract solution or the ruptured pericarp oil and then assessed the fruit appearance during storage. As shown in Figure 5, after 7 d of treatment, the color of the fruit epidermis coated with oil extract solution had faded (Figure 5A,B, second row), while the fruit epidermis with ruptured oil had turned brownish (Figure 5A, third row) and begun to rot after 10 d of treatment (Figure 5B, third row). In addition, the average water loss and cumulative water loss in fruit coated with the oil extract and ruptured pericarp oil were higher than those in untreated fruit (Figure 5C,D).

### 3.2. Water Loss

Fruit water loss seriously affects fruit quality. The average water loss of fruit at 26 °C was 7.2-fold higher than that of fruit at 5 °C from 0–45 d of storage (Figure 6A). After 105 d of storage at 26 °C, the accumulated water loss per fruit accounted for 12.8 ± 1.4% of the original weight, while that of fruit stored at 5 °C was only 2.4 ± 0.2% (Figure 6B). Water loss of fruit stored at different temperatures (5, 15, and 22 °C) showed that the higher the temperature, the greater the water loss per fruit during storage (Figure 7, Appendix A).

In addition to low temperature storage, the application of different light also affected fruits’ physiological metabolism. To test the effect of light on water loss, continuous illumination with five LED lights were used for 21 d. Blue light and red light obviously decreased the amount of water loss of navel orange when compared with white light (Figure 8B); similar changes in the average water loss per fruit were observed during storage (Appendix A). Low-temperature plasma also inhibited water loss of navel orange fruit during storage (Figure 9). The accumulated water loss from navel orange fruit treated with low-temperature plasma for 3 or 6 min was lower than that of untreated fruit after 7 d of storage at 22 °C (Figure 9C), but no obvious difference in color was observed (Figure 9A,B). A similar change in the average water loss per day was also observed (Appendix A).

### 3.3. Fruit Quality

Water loss caused a decrease in fruit firmness. In this study, we observed that the lower the temperature, the higher the fruit firmness, with a significant difference in the firmness of fruits stored at 5 and 22 °C (Figure 10B) and a close association between the firmness and the water loss at 5 °C (Figure 7). Furthermore, a significant difference existed in TA content of fruit stored at 5 °C and 15 °C when compared with those stored at 22 °C (Figure 10D), but there was no significant difference in TSS content (Figure 10C), which indicated that the TA content was more responsive to temperature at the early storage stage. Storage at 5 °C decreased the TSS content at the early storage stage but decreased the TA content at the late storage stage, while storage at 26 °C resulted in a decreased TSS content at the late storage stage and a reduced TA content at the early storage stage (Figure 11). In general, TSS content was dependent on temperature at the second storage stage (30–45 d), while TA content was dependent on temperature at the first storage stage (1–30 d) and the third storage stage (45–90 d).

Light and ionization can be used as green preservation technologies. The quality of postharvest fruit needs to be ensured. The fruit firmness and TSS and TA contents were not significantly different between the light-treated and control fruit (Figure 12). Similar results were observed for the TSS and TA contents of navel orange fruit after low-temperature plasma treatment (Table 1). These results indicated that the fruit quality could not be affected by a short-time treatment using light or low-temperature plasma.

## 4. Discussion

Fruit deterioration during harvesting, transport, and storage is an unavoidable issue that causes economical losses for farmers and enterprise. Low-temperature storage as a common method is used widely to delay fruit senescence and maintain fruit quality [24]. An alternative common preservation technology is the use of chemical fungicides [1,3,4,5,6]. However, increasing resistance of pathogenic fungi to chemical fungicides has emerged; for example, the major facilitator superfamily transporter in *P. digitatum* can confer resistance to fungicides [25]. Protein O-mannosyltransferases (Pmts) were responsible for the morphogenesis and virulence of pathogenic fungi, while mutation of Pdpmt2 was found to increase tolerance to the antifungal peptide PAF26 in *P. digitatum* [26], which indicated the pathogen can develop drug resistance through gene mutation under long-term application of fungicides to postharvest fruit. In addition, chemical fungicides applied to confer the resistance against pathogens must be considered. For example, the half-lives of the broad-spectrum fungicides imazalil and o-phenylphenol were 15–18 and 15 d, respectively, while the level of pyrimethanil was nearly unchanged within 28 d after application [27]. Therefore, physical technologies based on temperature, light illumination, and low-temperature plasma with a high sterilization efficiency without chemical residues should be developed for preserving fruit intended for human consumption. The study elucidated the biological basis and possibility of combining these treatments for green preservation.

In a previous study, most molds were found to cease their biological activities after treatment for 1 h at 50 °C [13]. Heat shock treatment can induce the production of reactive oxygen species and lignin biosynthesis in fruit and, therefore, promote disease resistance [16] and reduce respiration rate [15,28]. Application of LED blue light was found to induce the synthesis of phenolic compounds in citrus, resulting in a resistance to molds [18], while red and blue light could increase the content of capsaicinoids for preserving the postharvest commercial and nutritional quality of chili pepper (*Capsicum annum* L.) [29]. In addition, blue light can reduce the vitality of mold and impair its ability to infect fruit [19]. Treatment with UV-C at 180–280 nm and 10.5 kJ m^−2^ was found to maintain low ethylene production and respiration rate in lime fruit after 28 d of storage [30], while application of an alternating magnetic field inhibited the respiration of banana fruit [31]. These studies have confirmed the biological effects of physical handling technologies on postharvest fruit during storage, but the underlying physiological and metabolic mechanisms of fruits’ response to physical treatments need to be clarified in detail.

Fruit water loss is related to the physiological activity of postharvest navel orange in long-term storage. Our results illustrated that the water loss rate of orange fruit at the early stage of storage at 26 °C was relatively high (Figure 6) and that exocarp played an important role in water loss control (Figure 5). Nutrients and water would be transported to the peel, while energy consumption of the peel could cope with external biotic and abiotic stresses to maintain fruit freshness [32]. At different stages of storage, epidermal wax can boost the growth of *P. digitatum* hyphae, whereas cutin can inhibit conidial germination [33]. Therefore, it is necessary to study the relationship between peel composition and fruit rot in long-term storage. The improved disease tolerance of the navel orange was mainly dependent on jasmonic acid biosynthesis, downstream resistance gene expression, and metabolite accumulation [34]. For example, application of the synthetic auxin 2,4-D delayed fruit senescence by changing the levels of some endogenous hormones and water status, up-regulating the expressions of defense-related genes and improving the stress defense capability [35]. Recent investigations have shown that the relative humidity in the storage environment greatly affects citrus quality by influencing water loss, and the changes in aquaporin expression may account for the differences in the water loss rate and water transportation in the peel and pulp of citrus during storage [36]. Storage time and temperature also greatly affected water status, shelf-life, and quality of navel orange fruit [37]. In our study, the pattern of water loss from navel orange fruit in relation to TSS and TA metabolism during long-term storage could be further elucidated based on carbohydrate metabolism and tricarboxylic acid cycle and the transport of water by aquaporin.

## 5. Conclusions

In this study, we found that navel orange fruit began to rot after 90 d of storage at 5 °C or after 20–45 d at 26 °C. The water loss rate of the fruit at the early stage of storage at 26 °C was relatively higher when compared with that at 5 °C. Water loss of postharvest navel oranges was closely associated with the physiological activity, as shown in the different patterns of the reduced TSS and TA contents during long-term storage. Applications of low-temperature plasma, blue light, and red light inhibited water loss and reduced rot of the navel orange. Further studies are necessary to determine how to use low-temperature plasma to control pathogens in a short treatment time and reduce the potential unwanted effects on peel composition [38], answer the question of why red and blue light inhibit these physiological activities, and determine how the interaction of different storage stages and temperatures affect the metabolism of TSS and TA of navel orange fruit during storage. The relationship between metabolism and water loss should also be emphasized. Taken together, our findings indicated that a green technology for preserving navel orange fruit during storage could include low-temperature plasma, followed by a suitable low-temperature storage under continuous red or blue light illumination.

## Figures and Tables

**Figure 1 foods-11-03707-f001:**
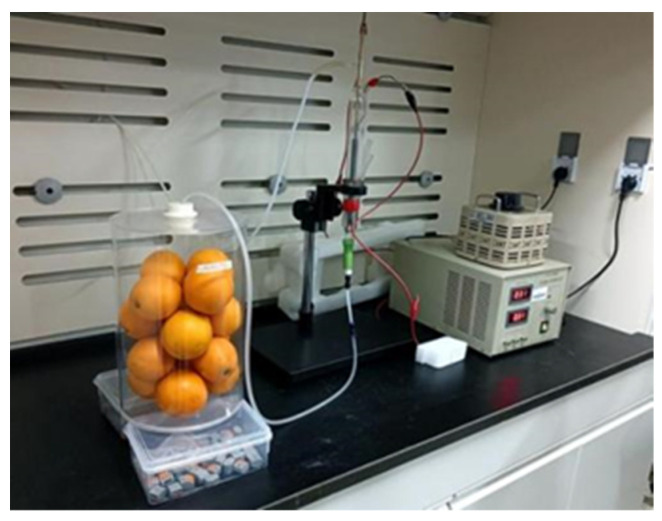
Low-temperature plasma produced by air and oxygen ionization.

**Figure 2 foods-11-03707-f002:**
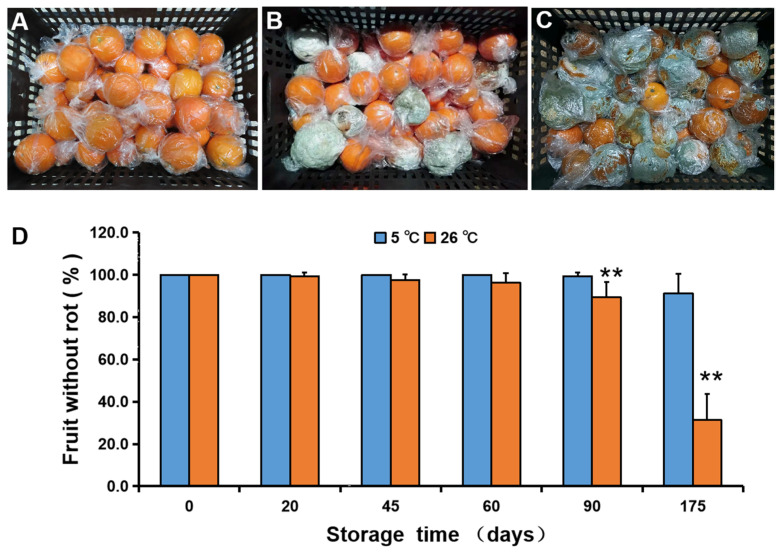
Fruit without rot in navel oranges during storage at 26 and 5 °C. (**A**–**C**) Images of bagged fruit at the beginning of storage (**A**) and after 6 months (**B**) and 1 year (**C**) at 5 °C storage condition. (**D**) Percentage (%) of bagged fruit without rot at 26 °C (blue) and 5 °C (orange) storage condition. The error bars represent the mean ± SD. **, *p* < 0.01 (*n* = 20).

**Figure 3 foods-11-03707-f003:**
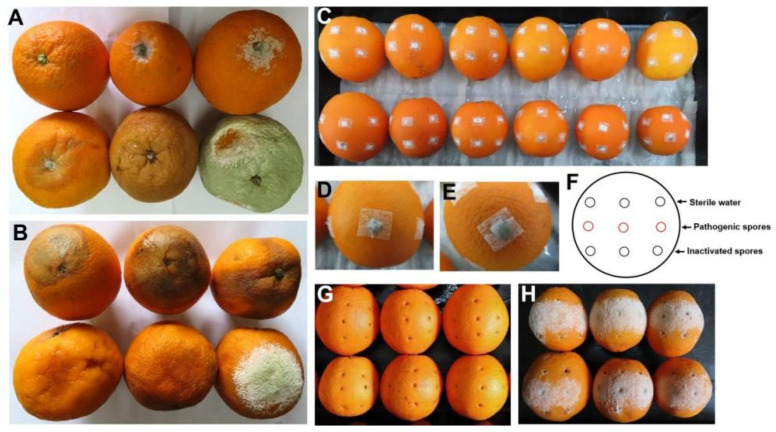
Fruit rot after 6 months of storage at 5 °C and after in vivo inoculation of pathogenic spores. (**A**). During storage, the mold slowly infected the fruit pedicle and caused the entire fruit to rot after 6 months of storage at 5 °C. (**B**) Mechanical damage on the fruits’ surface resulted in rot after 6 months of storage at 5 °C. (**C**) Noninvasive inoculation of the fruit surface with filter paper containing pathogen spore solution. (**D**,**E**) Enlarged view of the fruit with moldy patches. (**F**) Diagram of in vivo drilling inoculation of fruit surface. The first row of holes was inoculated with sterile water (negative control) and had no decayed phenotype. The second row was inoculated with untreated pathogenic spores (positive control), and the third row was inoculated with a spore suspension of pathogen inactivated by 10 min treatment of low-temperature plasma produced by the ionization of oxygen. (**G**) Initial phenotype of the fruit after in vivo drilling inoculation. (**H**) Fruit rot in the second row after 3 d of drill damage inoculation at 26 °C.

**Figure 4 foods-11-03707-f004:**
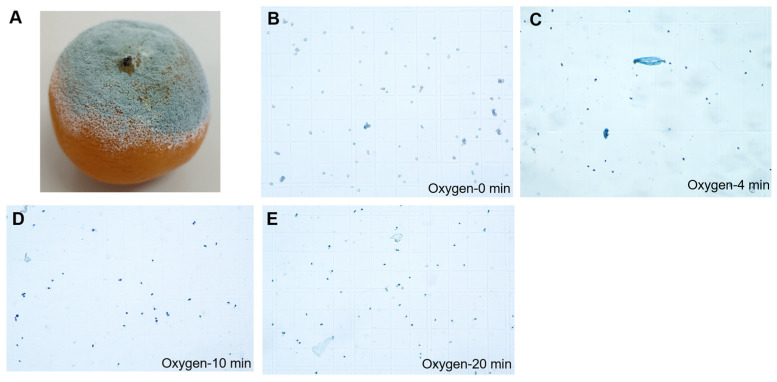
Effect of treatment with low-temperature plasma (oxygen ionization) on pathogenic spores. Appearance of a rotten fruit (**A**). Appearance of stained spores after 0 min (**B**), 4 min (**C**), 10 min (**D**), and 20 min (**E**) of low-temperature plasma treatment. Pathogenic spores were dyed with trypan blue.

**Figure 5 foods-11-03707-f005:**
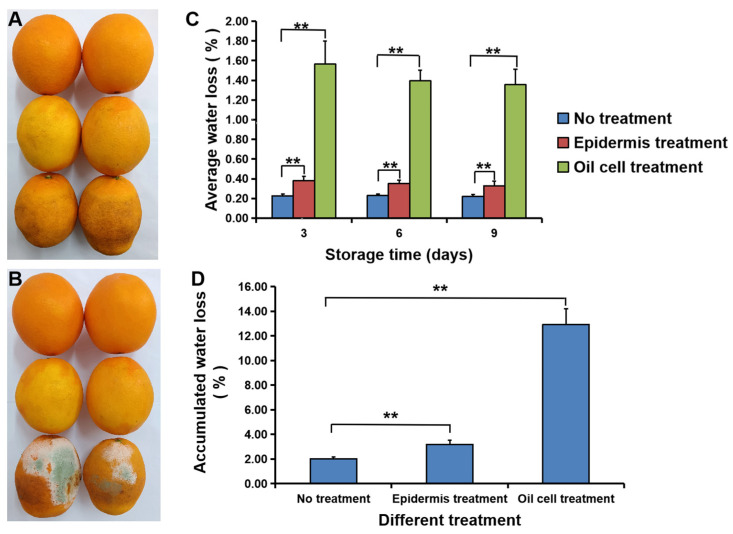
Disruption of pericarp oil cells increased fruit rot and water loss. (**A**,**B**) the phenotypes of untreated fruit (top), the fruit treated with the oil extract solution (middle) and ruptured pericarp oil (bottom) after 7 d (**A**) and 10 d (**B**) of treatment. (**C**,**D**) The average water loss per day (**C**) and cumulative water loss (**D**) in untreated fruit, and the fruit treated with the oil extract solution and ruptured pericarp oil. The error bars represent the mean ± SD. **, *p* < 0.01 (*n* = 6).

**Figure 6 foods-11-03707-f006:**
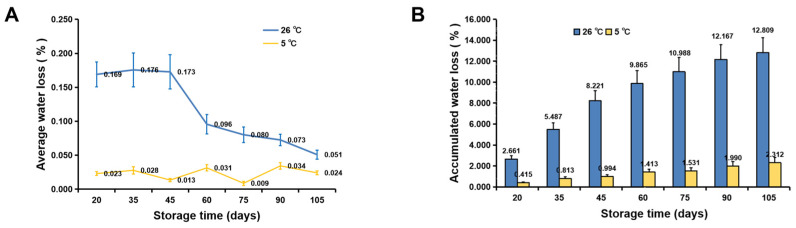
Water loss of navel orange fruit during storage at 26 and 5 °C. (**A**) Average water loss per fruit. (**B**) Accumulated water loss per fruit. The error bars represent the mean ± SD (*n* = 20).

**Figure 7 foods-11-03707-f007:**
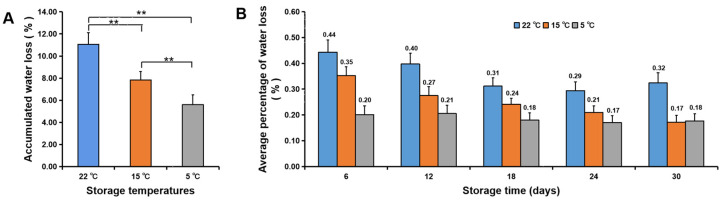
Water loss of navel orange fruit during storage at 5, 15 and 22 °C. (**A**,**B**), Accumulated water loss per fruit (**A**) and average water loss per fruit (**B**). The error bars represent the mean ± SD (*n* = 20). **, *p* < 0.01.

**Figure 8 foods-11-03707-f008:**
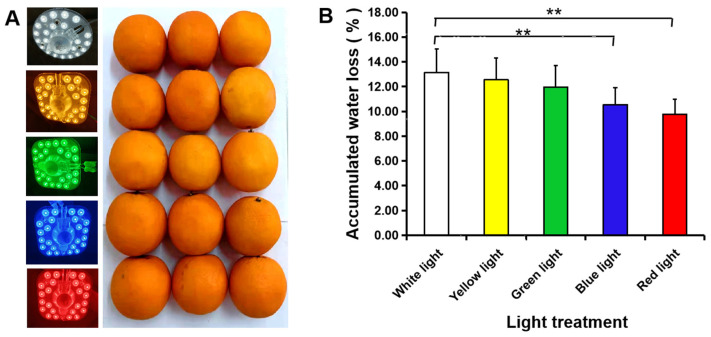
Illumination with blue or red LED light inhibited water loss in navel orange fruit. (**A**) Fruit phenotypes (right) after 21 d of different LED light treatments (left). (**B**) The accumulated water loss per fruit after 21 d of light illumination. The error bars represent the mean ± SD (*n* = 10). **, *p* < 0.01.

**Figure 9 foods-11-03707-f009:**
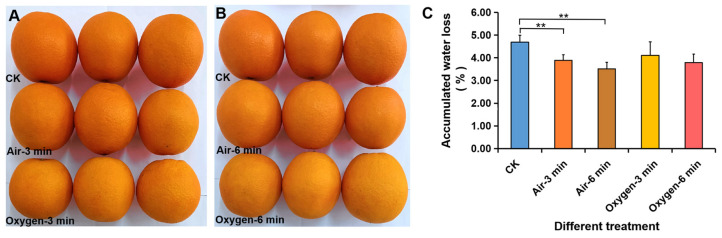
Inhibition of water loss of navel orange fruit by low-temperature plasma treatment. (**A**,**B**) The phenotypes of navel oranges without treatment (CK) and treated with low-temperature plasma produced by ionization of air and oxygen for 3 min (**A**) or 6 min (**B**). (**C**) The total water loss per fruit for untreated navel oranges (CK) and treatment with low-temperature plasma produced by ionization of air or oxygen for 3 or 6 min. The error bars represent the mean ± SD (*n* = 10). **, *p* < 0.01.

**Figure 10 foods-11-03707-f010:**
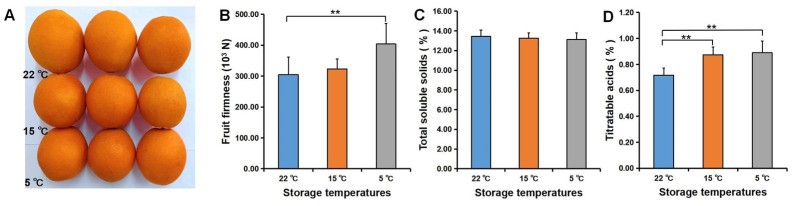
The firmness, total soluble solids, and titratable acids of navel orange fruit during storage at different temperatures (5, 15, and 22 °C). (**A**–**D**) Fruit phenotypes (**A**), firmness (**B**), total soluble solids (**C**), and titratable acids (**D**) of navel orange. The error bars represent the mean ± SD (*n* = 6). **, *p* < 0.01.

**Figure 11 foods-11-03707-f011:**
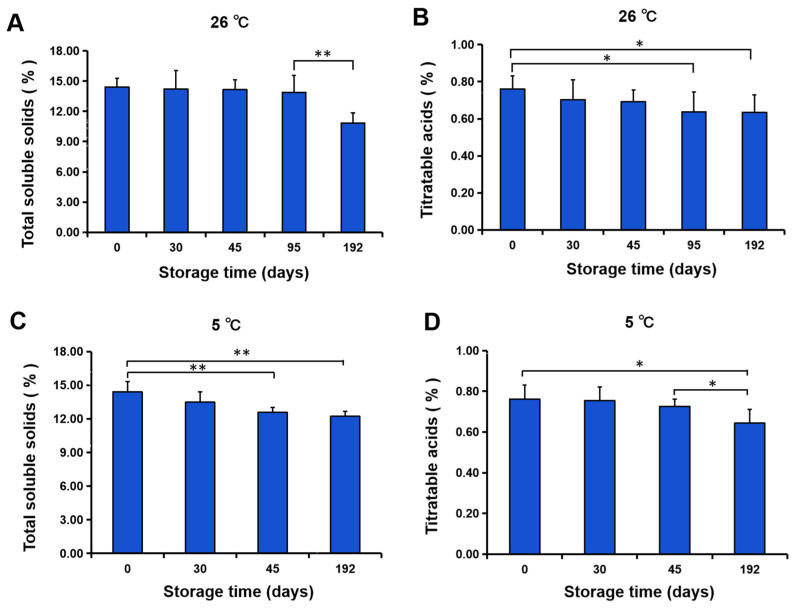
The changes in TSS and TA contents of navel orange fruit during storage at 26 and 5 °C. (**A**,**B**) Total soluble solids (**A**) and titratable acids (**B**) of fruit stored at 26 °C. (**C**,**D**) Total soluble solids (**C**) and titratable acids (**D**) of fruit stored at 5 °C. The error bars represent the mean ± SD (*n* = 6). **, *p* < 0.01; *, *p* < 0.05.

**Figure 12 foods-11-03707-f012:**
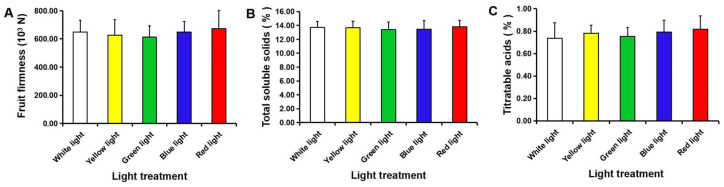
The fruit firmness (**A**), total soluble solids (**B**), and titratable acids (**C**) after 21 d of illumination with white, yellow, green, blue, or red LED light. The error bars represent the mean ± SD (*n* = 10).

**Table 1 foods-11-03707-t001:** The fruit quality after low-temperature plasma treatment ^1^.

Sample	Total Soluble Solids					Titratable Acids				
	Control	Air		Oxygen		Control	Air		Oxygen	
	0 min	3 min	6 min	3 min	6 min	0 min	3 min	6 min	3 min	6 min
1	10.8	13.3	11.9	11.9	10.2	0.67	0.91	0.76	0.73	0.62
2	13.3	13.4	12.8	11.8	11.8	0.81	0.95	0.86	0.74	0.73
3	12.6	10.6	12.5	11.6	10.7	0.75	0.57	0.6	0.64	0.68
4	13.0	11.9	11.4	11.9	13.1	0.73	0.74	0.53	0.65	0.71
5	11.6	12.0	11.0	11.7	11.2	0.85	0.76	0.67	0.75	0.65
6	12.1	11.6	12.2	10.8	12.4	0.77	0.65	0.67	0.69	0.72
Mean	12.23	12.13	11.97	11.62	11.57	0.76	0.76	0.68	0.70	0.69

^1^ The low-temperature plasma was generated from the ionization of air or oxygen for 3 and 6 min. After treatment, fruits were stored for 5 months at 5 °C in an air-conditioned room.

## Data Availability

All raw data used for figure and table generation in this manuscript can be obtained by contacting the corresponding author.

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
