# Peer review of "The Effects of Storage Temperature, Light Illumination, and Low-Temperature Plasma on Fruit Rot and Change in Quality of Postharvest Gannan Navel Oranges"

_foods, 2022, doi:10.3390/foods11223707_

Round 1

Reviewer 1 Report

Comments:

Experiment designs:

1.       Please briefly explain why you had two experiments for storage temperatures – 5 °C vs. 26 °C, and 5 °C vs. 15 °C vs. 22 °C.

2.       Light illumination and low-temperature plasma experiments showed their potential to reduce weight loss. Do you expect any residual effects after the treatments? Please provide some discussion in possible.

Statistics: Figures 8-10: you showed the significance in the accumulated water loss plots, but not in the average water loss plots. However, in the text, you described the significant differences in average water loss as well (blue and red light, and air-3 min and air-6 min treatments). The statements must be supported by data (tables and/or figures). In Table 1, you even did not show the control dada. In Figure 5, you also did not provide a photo for the control.

Writing:

1.       Grammatical issues: there are many writing issues, and I only list some examples.

Lines 2-4, title: please use the plural in “oranges”, or use “orange fruit” here and throughout the entire manuscript including tables and figures.

Lines 253-254: This is not a complete sentence. Also “associated” does not match “tightly”. Please replace “tightly” by “closely”.

2.       Remove the excessive words or information and clearly describe the facts. Such as:

Lines 12-15, excessive information: “economically important citrus”; ambiguous logic: “…water loss and quality change in relation to decay during storage”.

Lines 17-21, the sentence should be shorten to half. You may break the sentence if it is too long.

3.       Reduce simple replication in writing. For example, the Figure 5 title could be shortened to: “Effects of treatment time of low-temperature plasma (oxygen ionization) on pathogenic spores. A: appearance of a rotten fruit; B: 0 min (control); C: 4 min; D: 10 min and E: 20 min.  Pathogenic spores were dyed with trypan blue, which indicate the dead cells, as the living cells were not stained.” (You have to show the control!).

Others:

Figure 1. Should be deleted – it is a common procedure (it is ok to leave it in the supplementary).

Line 79: “numbered” to “labeled”.

Figures 6-10: Should remove all “per fruit” in the y-axis.  For average dada, please add “per day”; and for accumulated data, please add the total period into the x-axis.

Reviewer 2 Report

The manuscript is written with clear understanding of the project addressed. However, there are some concerns that need to be addressed to enhance the quality of the manuscript. My specific comments are as follows:

Abstract:

Elaborate more on methods of your study.

Introduction:

Based on your objectives, please compare how your study is different from those that have already been published

Materials and methods:

How many orange samples used?

“These experiments at two temperatures (5 and 26 ) from November 2020 to June 2021 and at three temperatures (5, 15 and 22 ) from November 2021 to March 2022 in air-conditioned rooms were conducted.” Why different temperatures are used form these two batches of fruit samples?

20 fruits for calculating water loss, then 6 fruits for quality measurement? Justify

Elaborate more on light treatment measurement

For low-temperature plasma, samples stored at 5 C and 22 C are selected. How about sample stored in 26 C?

Rupture of pericarp oil cells are evaluated at 22 C? How about other storage temperatures?

Results and discussion:

“Gannan’ navel orange begun to rot after 20 d of storage at 26 , but after 90 d at 5 , which showed that low temperature storage inhibited pathogen infection and reduced fruit rot (Figure 3).” The bar chart clearly showed no. of mildewed fruit in decreasing order, yet the authors claim that began to rot after 20 d of storage at 26 C? Please revise

For water loss, specify the values of water loss along the storage days (Fig. 7)

Similar with Fig. 8. Specify the values

Fig 9b and c are quite similar. Choose one between those two

Similar with Fig. 10, delete either 10c or d

“In addition, the average water loss and cumulative water loss of fruit coated with oil cell fluid and those with broken oil cells on the epidermis were higher than in untreated fruit (Figure 6C and 6D).” Move this sentence to its respective paragraph

Elaborate the results/values in Fig. 11-for firmness, TSS, TA

Combine Fig. 12 (a) and (c) into combined bar chart to show comparison between two temperatures. Similar with Fig. 12 (b) and (d)

Elaborate results for Figure 13 and Table 1. Discuss the difference between those plasma treatment

The findings lack in terms of justification and major findings.

Conclusions:

Add recommendation for future studies

General comments:

Please check the reference styles and grammar of the manuscript.

Author Response

Dear Reviewer

Round 2

Reviewer 1 Report

Please carefully debug the manuscript grammatically and professionally. I suggest you get a language service before submission. I will take some examples from the introduction:  

Lines 30-32: It is confusing which cultivar you used: Newhall or Gannan?

Line 33: “postharvest losses occur seriously with increasing storage” what is the meaning?

Lines 59-60: “The interaction between peel and mold is an important pathway in postharvest fruit rot”. I do not understand the logic. 

Author Response

Dear reviewer

According to your suggestions, I have carefully revised the manuscript and send the manuscript for English revision during the past ten days.  

The following is response for your some comments on the introduction of manuscript, and the revision of the manuscript is not only limited to the following issue:  

Lines 30-32: It is confusing which cultivar you used: Newhall or Gannan?

Response:’Gannan’ navel oranges indicate navel oranges (Citrus sinensis Osbeck cv. Newhall), which are planted in Ganzhou City in southern of Jiangxi Province, so ’Newhall’ is the cultivar.

 I have revised the sentence in “1. Introduction” to “’Gannan’ navel oranges (Citrus sinensis Osbeck cv. Newhall), which are planted in Jiangxi Province in southern China”.

revised the sentence in “2.1. Plant Materials” to “Fruit of ‘Gannan,’ navel orange, Citrus sinensis Osbeck cv. Newhall, were obtained from an orchard in Xinfeng County, Ganzhou City”

Line 33: “postharvest losses occur seriously with increasing storage”what is the meaning?

Response:I have revised the sentence to “postharvest losses occur seriously as the fruit rot”

Lines 59-60: “The interaction between peel and mold is an important pathway in postharvest fruit rot”. I do not understand the logic.

Response: I have revised the sentence to “The fruit peel is the initial site of the interaction between fruit and mold for postharvest fruit rot.”

Best wishes

Zhiqiang Gao

Reviewer 2 Report

The authors have addressed the comments. Hence, the paper can be accepted.

Author Response

Dear reviewer

According to your suggestions, I have carefully revised the manuscript and send the manuscript for English revision during the past ten days.

Best wishes

Zhiqiang Gao